



# A new method for calibrating marine biota living-depth using the 2016 Kaikōura Earthquake uplift

Catherine Reid[1], John Begg[2], Vasiliki Mouslopoulou[3,4], Onno Oncken[4], Andrew Nicol[1], Sofia-Katerina Kufner[4].

[1]Department of Geological Sciences, University of Canterbury, Private Bag 4800, Christchurch 8140, New Zealand.
[2]GNS Science, PO Box 30-368, Lower Hutt, New Zealand
[3]National Observatory of Athens, Institute of Geodynamics, Lofos Nimfon, Athens, 11810, Greece
[4]GFZ Helmholtz Centre Potsdam, German Research Centre for Geosciences, Telegrafenberg 14473, Potsdam Germany

*Correspondence to*: Vasiliki Mouslopoulou (vasiliki.mouslopoulou@noa.gr)

**Abstract.** The 2016 $M_w$ 7.8 Kaikōura Earthquake (South Island, New Zealand) caused widespread complex ground deformation including significant coastal uplift of rocky shorelines. This coastal deformation is used here to develop a new methodology, in which intertidal marine biota have been calibrated against tide-gauge records to quantitatively constrain pre-deformation biota living depths relative to sea level. This living depth is then applied to biologically measured tectonic uplift at three other locations along the Kaikōura coast. We also test how tectonic uplift measured using this calibrated marine biota compares to vertical deformation measured, at the same localities, using instrumental methods [Light Detection and Ranging (LiDAR) and strong motion data], and non-calibrated biological methods. Data show that where biological data is collected by RTK-GNSS in sheltered locations, this new tide-gauge calibration method estimates tectonic uplift with an accuracy of +/- ≤0.07 m in the vicinity of the tide-gauge, and an overall mean accuracy of +/- 0.10 m or 10% compared to differential LiDAR methods for all locations. Sites exposed to high wave wash, or data collected by tape-measure, are more likely to show higher uplift results. Tectonic uplift estimates derived using predictive tidal charts produce overall higher uplift estimates in comparison to tide-gauge calibrated and instrumental methods, with mean uplift results 0.21 m or 20% higher than LiDAR results. This low-tech methodology can, however, produce uplift results that are broadly consistent with instrumental methodologies and might be applied with confidence in remote locations where satellite data or local tide-gauge measurements are not available.



# 1 Introduction

Vertical displacement has been measured globally using inter-tidal marine biota on rocky coastlines which often provide important constraints for incremental uplift during large-magnitude earthquakes and cumulative geological uplift (Alaska: Plafker, 1965; California: Carver et al., 1994; Mexico: Bodin and Klinger, 1986; Ramirez Herrera and Orozco 2002; Costa Rica: Plafker and Ward 1992; Chile: Fitzroy, 1839; Castilla, 1988; Castilla et al., 2010; Farias, 2010; Vargas et al., 2011; Melnick et al., 2012; Argentina: Ortlieb et al., 1996; Eastern Mediterranean: Pirazzoli et al., 1982; Stiros et al., 1992;

Laborel and Laborel-Dugeun, 1994; Mouslopoulou et al., 2015a; Japan: Pirazzoli et al., 1985 and New Zealand: Clark et al., 2011; Mouslopoulou et al., 2019). Biological data offered the first written records of coastal uplift following earthquakes along the Chilean coast (Graham, 1824; Fitzroy, 1839; Wesson, 2017) and continue to provide important constraints for elastic rebound and coseismic slip processes together with the locations, depth and dip of causal faults (e.g., Melnick et al., 2012; Wesson et al., 2015; Mouslopoulou et al., 2015b; 2019).


Biological indicators such as lithophagid borings and stranded bioconstructions of corals, coralline algae and barnacles, along with brown algae, gastropods, bivalves, and additional intertidal species with locally reliable tidal elevation zones, have been used to estimate eustatic sea-level changes and rock uplift (or subsidence) due to tectonic processes (Laborel and Laborel-Dugeun, 1994). Quantifying earthquake uplift from such biological datasets has been achieved using a variety of

techniques, from simple measuring devices, such as tape measures, to laser survey methods and Global Navigation Satellite System (GNSS) techniques. While some studies (e.g., Melnick et al., 2012; Jaramillo et al., 2017) have successfully compared the reliability of the conventionally acquired biological uplift records against Real Time Kinematics (RTK) GNSS measurements, none have attempted to numerically and independently quantify the living depth of each biological marker. Jaramillo et al. (2017) compare pre- and post-deformation intertidal biota, but most studies, including this one, rely on post-

deformation data only. Clark et al. (2017) and Mouslopoulou et al., (2019) use a variety of methods to record deformation immediately following the Kaikōura Earthquake, however, their marine-biota measurements have not been calibrated. Moreover, none of the above studies have systematically compared the manually collected tape-measurements of coseismic uplift with instrumental earthquake-uplift datasets at individual localities to quantitatively assess the potential uncertainty inherent in the various techniques.


In this paper we use uplift produced by the November 14th, 2016 7.8Mw Kaikōura Earthquake (South Island, New Zealand) to develop a methodology for measuring coastal deformation utilising the vertical displacement of biozones. Capitalising on the long-term, continuous, high-precision tide-gauge readings at Kaikōura Peninsula, biological indicators of the intertidal zone uplifted during the earthquake are here utilised to: a) develop a new methodology with which to independently

calculate (and thus calibrate) the living depth of individual intertidal (algal) taxa (organisms which are widely used to measure coseismic vertical displacement), and; b) compare, at individual localities, the conventional biologically constrained



hand-held measurements of coseismic uplift to estimates acquired by various real-time remote sensing and other instrumental techniques, such as RTK-GNSS, LiDAR and strong-motion seismometers. Results may have application to inform future studies of the reliability of biological uplift measurements along rocky shores arising from large earthquakes at
mid-latitudes (particularly in the Southern hemisphere) and with moderate tidal ranges (e.g., ~2 m), especially where instrumental technologies, such as differential LiDAR, are not available.

## 2 Geological and biological setting

### 2.1 The 2016 Kaikōura Earthquake

The 2016 $M_w$ 7.8 Kaikōura Earthquake ruptured across the southern end of the Hikurangi subduction margin in northeastern
South Island of New Zealand (Mouslopoulou et al., 2019). Northeast of the Kaikōura Earthquake surface rupture, the plate boundary is dominated by oblique subduction of the Pacific Plate beneath the Australian Plate at rates of 40-47 mm/yr (De Mets et al., 1994) (Fig. 1, inset). At the southern termination of the subduction, relative plate motion is transferred onto the Alpine Fault via strike-slip on the Marlborough Fault System (MFS) (Pondard and Barnes, 2010; Wallace et al., 2012). The MFS generally strikes parallel to the relative plate motion vector and these active faults mainly accommodate right-lateral
strike-slip with the amount of fault-related uplift increasing towards the coast. Offshore and east of the surface rupture, plate boundary deformation manifests itself as an accretionary prism complex. The accretionary complex and eastern MFS are underlain by the Pacific plate which, based on the presence of a Wadati-Benioff Zone, extends to a depth of at least 200 km beneath the northern South Island (Eberhart-Phillips and Bannister, 2010). The subducting slab is at a depth of ~20-30 km beneath the faults (Nicol et al., 2018) and ruptured in response to slip triggered by the surface-breaking faults during the
earthquake (Mouslopoulou et al., 2019).

The Kaikōura Earthquake is the largest ($M_w$=7.8) historic earthquake to have ruptured within the southern termination of the Hikurangi subduction margin (Mouslopoulou et al., 2019). The earthquake comprised a complex network of at least 21 strike-slip, thrust and oblique-slip faults that ruptured the ground surface and straddle the coastline of the northeast South
Island (Clark et al., 2017; Hamling et al., 2017; Litchfield et al. 2018). The event's complexity is reflected in the moment tensor of the main shock which features only 65 to 75% double couple percentage (GEOFON http://geofon.gfz-potsdam.de accessed March 20, 2017) and is characterized by an oblique mechanism, with components of thrusting and right-lateral slip. Fault ruptures generally propagated northwards from the epicentre for about 200 km, with a focal depth of the main shock at 15 km (Hamling et al., 2017; Kaiser et al., 2017; Cesca et al., 2017). The resulting surface ruptures vary in strike from east-
west to north-northwest with faults having east-northeast strike being primarily right-lateral strike-slip and more northerly striking faults accommodating strike-slip and reverse displacement (Nicol et al., 2018). The earthquake ruptured three faults (Hundalee, Papatea and Kekerengu faults) that cross the coastline and locally produced differential uplift of the rocky





shorelines. Vertical displacement of -0.5 to +8 m occurred along >100 km of coastline with the highest values in the hanging

wall of the reverse sinistral Papatea Fault north of Kaikōura (Clark et al., 2017; Litchfield et al. 2018; Mouslopoulou et al.,

2019). The coastal section examined in this paper is crossed by the Hundalee Fault (Fig. 1; see also Figure 1c in

Mouslopoulou et al., 2019) which accommodated a component of reverse displacement and uplift of the coast up to ~2 m. In

addition to the mapped surface faults, the spatial extent of coastal uplift and the widespread occurrence of tsunamis, which

span distances of up to ~250 km from Kaikōura south (Power et al., 2017), suggest that faulting at the ground surface may

have been accompanied by slip on the subduction interface and an offshore thrust fault that splays from the plate-interface to

extend within the accretionary prism complex (e.g., Cesca et al., 2017; Mouslopoulou et al., 2019).

## 2.2 Biological Setting

The northern Canterbury coastline is predominantly exposed and strikes northeast-southwest, and is broken only by the

promontory of the Kaikōura Peninsula (Fig. 1). Hinterland topography is steep and the coastal strip is narrow, exposing

mainly bedrock beneath bouldery shorelines which are interrupted by bays with gravel-dominated beaches. Prevailing winds

from the northeast (summer months) and southwest (winter months) maintain year round exposure and the coastline supports

a biota adjusted to this high energy setting. The region is in a cool temperate oceanographic setting with diurnal tides. Daily

tidal variation is up to ~2 m, which in turn influences the living depths of intertidal biota.

The intertidal biota in this cool temperate setting is dominated by seaweeds, typically the large brown algae *Durvillaea*

*antarctica* (bull-kelp), *D. willana, Carpophyllum maschalocarpum* (Fig. 2a), and *Hormosira banksii,* coralline algae (Fig.

2b), barnacles and mobile invertebrates (Marsden, 1985). Encrusting invertebrates, such as mussels and oysters, are present

but not common on this stretch of coast. On the Kaikōura Peninsula species diversity is high, with up to 78 species present in

a single intertidal transect (Marsden, 1985). The vertical distribution of species on these rocky shores is controlled by

exposure as well as interspecies competition (Goldstien pers. comm., 2017). The rocky shores around Kaikōura support three

major biozones that approximately correspond to tidal height: a) an upper belt of littorinid gastropods (e.g., *Littorina*

*unifasciata* and *L. cincta*) and barnacles (e.g., *Epopella plicata*); b) a mid-tidal region dominated by grazing molluscs (e.g.,

*Cellana denticulata, Melagraphia aethiops* and *Turbo smaragdus*); and c) a lower zone of brown algae (e.g., *Durvillaea*

*antarctica* and *Carpophyllum maschalocarpum*) (Marsden, 1985). When the shoreline was inspected, about two and a half

months after earthquake uplift, many mobile taxa were absent and living or dead remains of stranded encrusting or attached

taxa, such as barnacles, coralline algae and brown algae, dominated the shoreline. The green alga *Ulva* is normally present in

limited amounts (Marsden, 1985), however, following the Kaikōura Earthquake and shoreline disturbance, growth of this

alga was prolific and it subsequently covered much of the post-earthquake intertidal zone in the study area (Fig. 2b-d). This

proliferation was accompanied by the death and bleaching of stranded coralline red algae forming a distinctive white crust on



rocky surfaces (Figs. 2b & d), which were often visible at kilometre-scale distances and were the most obvious visual indicator of uplift along the coastline.

In this study the brown algae *Durvillaea* and *Carpophyllum* are utilised to measure coastal uplift. *Durvillaea* is restricted to the southern hemisphere and occurs on rocky coastlines throughout New Zealand, while *Carpophyllum* is endemic (Adams, 1994). Around the Kaikōura Peninsula and north Canterbury coast, holdfasts of *Durvillaea antarctica* (bull-kelp) and *D. willana* (Fig. 2a), are anchored by a fleshy non-calcified holdfast to coralline encrusted rocky surfaces in the lower inter-tidal zone (Adams, 1994; Nelson, 2013) and holdfasts extend sub-tidally by 1-2 m. Individual plants have fronds 3-5 m in length that typically drape down from the inter-tidal zone to depths of ~5 m (Adams, 1994; Nelson, 2013). On sites exposed to higher wave action, holdfasts of *Durvillaea* may appear higher in the intertidal zone in response to increased wave wash (Marsden, 1985), however, in sheltered areas and sites where waves are baffled holdfasts may be exposed at spring low tides, but not at neap low tides (Goldstien pers. comm., 2017). By contrast, *Carpophyllum* is only present in the low intertidal zone where it forms a distinct band at low water (Nelson, 2013) (label *C* in Fig. 2c), and is not normally exposed at low spring tides (Goldstien pers. comm., 2017). Although both *Carpophyllum* and *Durvillaea* may be present on open coasts (Fig. 2a), *Durvillaea* dominates in exposed sites and *Carpophyllum* is more abundant at relatively sheltered locations. One or both of these brown algae were present at all the rocky coastal sites visited in this study, making *Carpophyllum* and *Durvillaea* an excellent combination of biozone markers for measuring coseismic uplift.

The reproductive season for *Durvillaea* is during the winter months peaking in August and harvesting studies have shown slow resettlement when fronds are removed in September through February (Hay and South, 1979). The intertidal zone on the Kaikōura coast is undergoing recovery from the November 2016 earthquake and stabilised intertidal zones are not yet re-established. In temperate climate settings this may take several years, as shown by Castilla and Oliva (1990) following the 1985 Chile earthquake.

## 3 Methods

To measure coseismic uplift due to the Kaikōura Earthquake, independent methods utilising marine biological sea-level indicators, tidal gauge measurements, remote sensing techniques (RTK-GNSS and LiDAR) and strong motion recordings are used. The characteristics of each dataset collected and the methodology used to derive tectonic uplift are presented below. All uplift data are available in the Supplementary Material.





## 3.1 Kaikōura Tide Gauge

New Zealand has 15 tide-gauges which record tidal variation, eustatic sea-level changes and vertical motion of the coast. The
Kaikōura Tide Gauge (Fig. 1) measures sea-level relative to two Druck PTX1830 sensors (KAIT 40 and 41 each referenced
to different datums) located at the end of the wharf at Kaikōura (WGS-84 -42.41288°, 173.70277°; NZTM 1657824,
5304141). In this study, data from the KAIT 41 sensor (http://apps.linz.govt.nz/ftp/sea_level_data/KAIT/) are used
exclusively to maintain internal consistency, however, results would be the same had KAIT 40 been used. The instrument is
fixed to bedrock beneath the wharf, referenced to nearby benchmarks, including one on the wharf itself (LINZ geodetic code
EEFL) and records sea-level at one minute intervals. The data are recorded in UTC time and the water-levels represent water
surface elevation above the base of the tide-gauge in metres. The tide-gauge was established in late May 2010 and operated
continuously through the period of the November 14th Kaikōura Earthquake recording tectonic uplift at the site. KAIT 41
Tide Gauge data assembled for this study spanned the period from December 1st, 2015 to February 7th, 2017 and indicate that
tidal range varies between a spring tide average of c. 2 m and c. 1.25 m during neap tides (Table 1). Spring low-tides before
the Kaikōura Earthquake registered c. 2.05 m on the gauge while spring high-tides were c. 4.05 m. After the earthquake,
low-water spring tide measured c. 1.1 m and high-water spring c. 3.1 m. Neap tides measured c. 2.5 m (low) and c. 3.7 m
(high) before the earthquake and c. 1.5 m (low) and c. 2.75 m (high) after the earthquake (Table 1).

To determine the absolute uplift value from the tide-gauge data ($U_{TG}$; see Suppl. File S1) we used the following
methodology: a) Subtracted the high-spring and high-neap tide readings before the earthquake from those after the
earthquake; b) Averaged high-tide and low-tide readings from several tidal cycles (3 day period) before and after the
earthquake; c) aligned pre-earthquake tidal data with post-earthquake data and incrementally adjusting them until a best fit;
d) compared the average water elevation from a pre-earthquake month to the same month's data after the earthquake (e.g.
December 2015 against December 2016); and e) calculated the difference in average waterline elevations for an extended
period (44 days) before and after the earthquake (Oct. 30th to Dec. 27th). The average uplift ($U_{TG}$) estimated from the above
steps (Table 1) is subsequently used to independently estimate the preferred dwelling range of the biological holdfast used in
this study (see Sect. 3.2). It has also acted as a reference point against which all other instrumental and hand-held
measurements are compared.

Some limitations on calculating vertical displacement from tide-gauge records arise from the specific circumstances
associated with the November 14th, 2016 $M_w$ 7.8 Kaikōura earthquake. This event struck during a period of sharply
increasing tidal change due to high spring tides (related to lunar perigee and approaching solar perihelion) that culminated a
few days after the earthquake. In addition, the earthquake generated a significant tsunami (Power et al., 2017), the effects of
which persist in the tide-gauge record for at least 12 hours after the earthquake. Further, a day after the earthquake, Kaikōura
was subjected to a southerly storm with powerful swells and these are also apparent in the tide-gauge data. These factors



result in some blurring in the precision of uplift data deriving from the difference between pre-earthquake and post-earthquake data.

## 3.2 Biological Data

Biological records of coseismic uplift were collected using the elevation of approximately 400 stranded algal holdfasts during a ten-day period, approximately two and a half months after the Kaikōura Earthquake (Suppl. File S2). Decay of attached and uplifted biota was well-advanced and, in most cases, uplifted remnants of marine algae, our primary target species, were restricted to holdfast stumps of *Durvillaea* or *Carpophyllum* with brittle fronds attached (Figs. 2b-d). Despite the decay of algae, the position of the remaining stumps clearly reflected pre-earthquake algae distribution evidenced by a

lack of rock "scarring" where removed stumps might also remove other intertidal biota and often expose fresh rock surfaces. The biological data presented in this paper were collected from close to the Kaikōura Tide Gauge on the northern side of the peninsula, Kaikōura Harbour on the south side, and from two localities along the south Kaikōura coastline, Paia Point and Omihi Point (Fig. 1).

At all localities uplift was apparent from the exposure and subsequent degradation of intertidal biota with algal holdfasts exposed above the waterline, and measurements were collected on rising or falling mid- and low-tides. Holdfasts were preferentially measured on rock faces sheltered from, but retaining connection to, the open sea, to minimise error introduction by the potentially higher tidal position of *Durvillaea* in wave-washed sites. Each site was visually assessed to establish the upper extent of holdfasts, and the uppermost holdfasts were measured (as they will be closest to the pre-

earthquake upper limit of each species). In sites with boulders rather than bedrock exposure, only boulders that showed a portion of their surface to have been clearly within the pre-earthquake mid- or upper-tidal zone (evidenced by bare or barnacle encrusted surfaces) were selected for measurement, therefore ensuring the upper limit of holdfasts were represented.

Two different methods were used to measure the vertically displaced biota. The primary method of collection of field data

was by Real Time Kinematic Global Navigation Satellite System (RTK GNSS). At each site the water-level was measured in the most sheltered area available to minimise wave effects, and the time the measurement was collected was recorded. Following measurement of the waterline, up to twenty holdfasts (either or both *Carpophyllum* and *Durvillaea*) were measured within close proximity. This RTK collection method did not require the waterline measurement site and the holdfasts to be immediately adjacent to each other. Additional biological data were collected using a tape-measure. Tape

measurements were collected between the waterline (measured between wavelet peaks and troughs) and the uppermost algal holdfasts on rock surfaces. Sheltered faces were again preferentially measured, however, the requirement to have stranded holdfasts immediately adjacent to waterline meant that sites exposed to wave-wash needed to be used to achieve





approximately twenty measurements. Each reading for both methods (RTK or tape) was annotated with the alga measured and relative site exposure (exposed or sheltered) and time of measurement recorded.


These field measurements of apparent uplift were then further processed to determine the total uplift, taking into account the time of data measurement and the pre-earthquake living position of algal holdfasts. Three different methods were used for calculating tectonic uplift from the vertical offsets of the biota. These were: a) tide-gauge calibration, b) NIWA tide-forecaster measurement and, c) LINZ tide-prediction charts. The first method utilised data from the Kaikōura Tide Gauge

and differs significantly from the two tide-prediction methods by calibration to real-time water-level records of the Kaikōura Tide Gauge. The NIWA forecaster and LINZ tidal chart methods are included, however, to simulate locations where real-time tide-gauges are not available. All data and calculations are presented in the Supplementary File S2.

### 3.2.1 Deriving a living-depth correction factor using the Kaikōura Tide Gauge ($X_{C/D/G}$)

This new method seeks to determine a living-depth for each species and positions the elevations of the stranded holdfasts relative to the pre-earthquake tidal cycle (Fig 3). The living-depth is described here by the correction factor $X_{C/D}$, which is treated as a constant for *Carpophyllum* ($X_C$), *Durvillaea* ($X_D$) or a combination of both ($X_G$), respectively. $X_{C/D/G}$ were determined by the Eq. (1):

$$X_{C/D/G} = ((H+OM_{C/D/G})-U_{TG})-MLWS \tag{1}$$

where H is the waterline height at the tide-gauge at the time of data collection (which can be accessed from http://www.linz.govt.nz/ and which was averaged here over 10 min intervals to mitigate local fluctuations); $OM_{C/D/G}$ is the observed height above the waterline of each stranded holdfast (determined by subtracting RTK waterline height measurement from each RTK holdfast measurement per site, or directly using tape-measurements; the indices C/D/G correspond to measurements for the different holdfasts); MLWS is the average tide-gauge reading for mean low water-spring tide (1.1 m for KAIT 41; see Table 1); $U_{TG}$ is this uplift calculated at the tide-gauge by the method described in Sect. 3.1.

As *Carpophyllum* and *Durvillaea* prefer slightly different living positions in the inter-tidal zone, $X_{C/D}$ was determined
separately for each species. A general correction factor $X_G$, using both *Carpophyllum* and *Durvillaea* holdfasts was also determined, to be applied at sites where holdfast species were not known or determined, or insufficient numbers of each were available and data were pooled by necessity. Further, only sheltered sites nearest the Kaikōura Tide Gauge were used to determine $X_{C/D/G}$. To calculate the correction factor, data were pooled by species irrespective of site. The method described here uses intertidal algae as marker species, as at Kaikōura these are readily available attached biota. However, locations
with any other attached inter-tidal biota with a restricted tidal range could be used to calculate this correction factor.



### 3.2.2 Deriving tectonic uplift using the Kaikōura Tide Gauge method ($U_{B(TG)}$)

Once the $X_{C/D/G}$ correction factor was derived as described above, coseismic uplift was calculated from biological data
pooled by site in the location studied, using Eq. (2):


$$U_{B(TG)} = ((H+OM)-MLWS)-X_{C/D/G} \tag{2}$$

### 3.2.3 Deriving tectonic uplift using the NIWA Tide Forecaster ($U_{B(NIWA)}$)

Uplift was also calculated from RTK data using tidal charts (https://www.niwa.co.nz/services/online-services/tide-forecaster)
that provide tidal predictions for sites between formal chart stations and attempt to account for local variation. For this
calculation, Eq. (3) is used:

$$U_{B(NIWA)} = (OM+H_{NIWA})-X_{C/D\_NIWA}, \tag{3}$$

where $X_{C/D\_NIWA}$ is a correction value (NIWA Forecaster calibrated correction), estimated to reflect the relative height of
*Carpophyllum* and *Durvillaea* within the tidal cycle. This value for X is independent of tidal-gauge data as used above and
relies on assessment of qualitative biological data only (Fig. 4). As described in Sect. 2.3, *Carpophyllum* in sheltered areas
with connection to the sea will not usually be exposed at low spring tide (LST) (Goldstien pers. comm., 2017). Tidal
prediction charts over one year were qualitatively assessed and a mean low spring tide height of 0.1 m ($X_{C\_NIWA}$) estimated
for the upper limit of *Carpophyllum* and used as the correction value for this species in data processing. Likewise *Durvillaea*
will be regularly exposed at low spring tides but usually not exposed at low neap tide (Goldstien pers. comm. 2017). A
correction ($X_{D\_NIWA}$) of 0.25 m was estimated, representing a regional height between spring and neap low tides. These
values for $X_{C/D\_NIWA}$ assume the height of *Carpophyllum* and *Durvillaea* are constant in both sheltered and exposed areas.

The value $H_{NIWA}$ was determined using the predicted tide heights and times from the NIWA Tide Forecaster website. The
NIWA Tide Forecaster provides tide height at user designated locations. $H_{NIWA}$ was calculated using the following Eq. (4)
from http://www.linz.govt.nz/

$$H_{NIWA} = h1 + (h2 - h1) [(cosA + 1)/2] \tag{4}$$




Where A= $\pi([(t - t1)/(t2 - t1)] + 1)$ radians and t1 and h1 denote the time and height of the tide (high or low) immediately preceding time t, and t2 and h2 denote the time and height of the tide (high or low) immediately following time t. Only time t is measured, t1 and t2 and h1 and h2 are derived from predictive tide charts.

### 3.2.4 Process to derive tectonic uplift using the LINZ tide prediction charts ($U_{B(LINZ)}$)

Land Information New Zealand (LINZ) tide charts available at http://www.linz.govt.nz/ provide fixed tide prediction charts for New Zealand primary and secondary ports and were also used to derive $H_{LINZ}$, using Eq. (3), and LINZ calibration correction values of 0.2 m for $X_{C\_LINZ}$ and 0.4 m for $X_{D\_LINZ}$ estimated as above from these charts. $H_{LINZ}$ was again determined by Eq. (4) defined above, and only RTK data was processed this way.

### 3.2.5 Sources of error

Data points collected by RTK GNSS were accurate to $\pm$ 5 cm, and this applies to both the waterline measurement at each site, and each holdfast measurement. Both of these measurements were used to derive OM, with a total error of $\pm$ 10 cm. Manually-collected biological data rely on the accuracy of the waterline measurement taken. While sheltered microsites were selected for these measurements, they were placed at an estimated median water-level between wavelets. This error is more pronounced when measuring waterline heights at exposed sites. Additionally the time at which the measurement was taken may have occurred when water-level was at either a positive or negative fluctuation from tidal prediction charts or Tide Gauge readings for sites south of Kaikōura. The total error is difficult to quantify, however, assessment of the Kaikōura Tide Gauge data show water-level fluctuations of less than +/- 0.1 m. Averaging tide-gauge data over 10 minutes helped mitigate the error resulting from the tide gauge itself, however, the error introduced by sea-level fluctuations away from the tide-gauge remained.

*Durvillaea* occurs at open coasts, however, at very exposed sites pre-earthquake holdfasts would have sat higher than average in response to increased wave wash and run-up. This potential error is difficult to quantify as deviation from average heights will be linked to wave heights and run-up at individual sites that may be modified following uplift. For this reason the most exposed sites were avoided (where possible) and data were collected from sheltered locations.





### 3.3 Remote sensing and strong motion uplift estimates

#### 3.3.1 Differential LiDAR ($U_{LiDAR}$)

Differential LiDAR has been developed along the coastal south Kaikōura region using pre- (DEM_Kaikōura_2012_1m) and post-earthquake (NZVD2016 and DEM_NZTA_1m) surveys of road and railway routes using a common geodetic datum for each survey. To minimise the impact of gravity-induced slope failures and horizontal tectonic displacement on sloping ground during the earthquake, the difference of the altitude of 1x1 pixels along the post-earthquake centreline of roads was used (Fig. 7). Specifically, for the Omihi Point and Paia Point study localities (see Fig. 1) the nearby State Highway-1 was

used, while for the Kaikōura Tide Gauge study-site, a section of the coastal road near the wharf that houses the gauge was used. The road sections that acted as a reference level have low relief (e.g., <10 cm relief) and are wider than the horizontal displacements recorded during the earthquake; thus, neither lateral tectonic displacement nor gravitational processes should significantly impact on the differential LiDAR measurements. Collectively, a total of 510 differential LiDAR points were collected and analysed (148 at the Kaikōura Tide Gauge, 152 points at Paia Point and 210 points at Omihi Point) (Suppl. File

S3). These data were used to produce mean uplift estimates of at each site with $2\sigma$ uncertainties of ±0.06-0.18 m (Table 6).

#### 3.3.2 Strong motion ($U_{SM}$).

A further independent instrumental uplift measurement by calculating the static vertical displacement recorded by the nearby strong-motion site KIKS (Fig. 1). The KIKS station is located 2.2 km south of the Kaikōura Tide Gauge (lat./long. -

42.426°N/173.682°E; NZTM: 1656161, 5302714; see Fig. 1) and operated by GeoNET. The Kinemetrics FBA-ES-T-BASALT 2420 sensor is located at 8 m elevation on the concrete floor of a single storey building at Kaikōura Harbour. Ground acceleration is recorded with a period of 0.005 s and data can be downloaded online from [ftp://ftp.geonet.org.nz/strong/processed/](ftp://ftp.geonet.org.nz/strong/processed/).

Static displacement was calculated from the vertical component of the instrument following the method of Wang et al. (2011) and using their software package *smbloc*, which applies an empirical baseline correction to remove linear pre- and post-event trends in the data. Static displacement derived with this method after large earthquakes has been shown to be robust (e.g. Schurr et al., 2012). Here, the resulting vertical displacement for the KIKS strong-motion station is 0.87±0.06 m (Table 6 & Suppl. File S4 for further details on data processing).

### 4 Results and comparison of methods

The Kaikōura Tide Gauge was co-seismically uplifted by 0.96 ± 0.02 m ($U_{TG}$) (Table 1 and 2) (see sect. 3.1) and represents a key reference point for this study. In addition to providing an independent estimate of uplift, the tide gauge data have been



used to calculate the living depth correction factor $X_{C/D/G}$ from all stranded biological holdfast data collected proximal to the Tide Gauge (Eq. 1) (Table 3) and test this method at the Kaikōura Tide Gauge (Fig. 5). Uplift estimates derived from direct
tide-gauge analysis (i.e., the method described in Sect. 3.1), are here compared to uplift estimates derived from biological methods (Equations 2 & 3 in Sect. 3.2) (Fig. 6a).

The calculated correction factors $X_{C/D}$ (Table 3) were applied to biological measurements collected proximal to the Kaikōura Tide Gauge (Fig. 5). RTK-GNSS survey data of *Durvillaea* and *Carpophyllum* for sheltered and exposed holdfasts produce
tectonic uplift values of 0.71 m to 1.13 m, with a mean of 0.97 m ± 0.08 m (Table 4, Fig. 5). Similarly, for all tape-measure data collected proximal to the tide gauge, tectonic uplift estimates range between 0.87 m and 1.35 m, with a mean of 1.05 ± 0.11 (Table 4, Fig. 5). The resulting analysis suggests *Carpophyllum* at sheltered sites recorded using RTK-GNSS and tape measure produce uplift estimates that are, within the uncertainties given, indistinguishable from uplift estimates based on the tide-gauge (0.96 m) and differential LiDAR (0.92 cm) (Fig. 5). By contrast, estimates of uplift using *Durvillaea* are always
higher than tide-gauge and differential LiDAR values. Tape-measurements of *Durvillaea* produced the highest biological uplift estimates with exposed *Durvillaea* recording a mean uplift of 1.21 m, which is 0.25-0.29 m above the tide-gauge and differential LiDAR values (Table 4, Fig. 5). These data suggest that *Durvillaea* should be regarded as providing maximum uplift estimates, supporting previous work in suggesting that *Durvillaea* at exposed sites should be used with caution (e.g., Clark et al., 2017).


The same biological data collected near the Kaikōura Tide Gauge was then grouped by data collection location (sets of approximately twenty data points) rather than holdfast type, and uplift estimates produced results of 0.99 m ± 0.07 m, 0.923 m ± 0.10 m and 0.98 m ± 0.07 m, while tape measures resulted in uplift estimates of 1.00 m ± 0.07 m, 1.12 m ± 0.11 m and 1.19 m ± 0.08 m, respectively (Fig. 6a, Table 5). In addition to directly measuring water-levels in the tide-gauge, the NIWA
Forecaster and LINZ tide-charts were used to calculate uplift in an effort to test the utility of tide-charts at remote locations where tide gauge and instrument data may not be available. These comparisons are illustrated in Table 5 and Figure 6. At the tide-gauge site, the LINZ Tide Chart produced, for *Carpohyllum,* uplift results 0.11-0.12 m greater than the tide-gauge method, while NIWA Forecaster chart produced uplift estimates of 0.04-0.05 m greater than the tide-gauge mean (Table 5). As was the case for the tide gauge calibration method, *Durvillaea* produced the greatest uplift at the tide gauge using the tide
chart method, with average uplift values of 1.18 m and 1.24 m. In summary, uplift estimates calculated from *Carpophyllum* holdfasts processed using the NIWA Forecaster tide charts (rather than LINZ charts), are the most similar to direct uplift of the tide-gauge itself, to the tide-gauge biological results and to LiDAR (plus 0-0.25 m), promoting their use in circumstances where a tide gauge is unavailable. LINZ tide chart methods produced results within 0.32 m of other methods.

To further test the utility of the Kaikōura calibration method, and the other methods under consideration, algae uplift data were also processed from the Kaikōura Harbour, Paia Point and Omihi Point sites. Data from these locations are not as





detailed as those collected at the tide-gauge study-site itself, with the distinction between *Carpophyllum* and *Durvillaea* or sheltered and exposed not always available.

At Paia Point, uplift estimates from all data collection and processing methods range from 1.12 m to 1.36 m, with a mean uplift of 1.24 m (Table 5; Fig. 6). While the biological uplift results are internally consistent, on average they are about 0.2 m higher than the differential LiDAR average uplift at this site, which is 1.05 m ±0.07 m (Table 6; Fig. 7). This higher estimate for biological data cannot be attributed to differences in species of algae or measurement technique, however, shoreline exposure cannot be excluded as a factor. The role of shoreline exposure may only be resolved once the uplifted shoreline is

recolonised with new *Carpophyllum* and *Durvillaea*. Algal uplift measurements collected at Omihi Point (Fig. 1), and processed using the tide-gauge calibration factor $X_{C/D}$ are within 0.07 m from one another and to uplift recorded by differential LiDAR (Tables 5 & 6, Fig. 6). RTK measurements from Omihi Point processed using the NIWA Forecaster and LINZ tide charts methods are 0.08 m and 0.23 m respectively above tide-gauge calibrated estimates. In summary, there is no systematic difference in the uplift estimates at Paia and Omihi Points between the different measurement techniques (RTK-

GPS vs tape-measure), species of algae (*Carpophyllum* or *Durvillaea*) or tide charts (NIWA Forecaster or LINZ tide chart) (Fig. 6).

At the Kaikōura Harbour site, where the KIKS seismic station is located (Fig. 1), uplift estimates from biological data, processed with the tide-gauge calibrated living depth methodology are 0.74 m ± 0.12 m, 0.85 m for NIWA calibrated

methods and 0.98 m for LINZ methods. These results bracket the uplift result recorded by the strong motion data of 0.87 m ±0.06 m (Fig. 6). At this locality, the comparison of the biological measurements with LiDAR has not been attempted as differential LiDAR data produced inconsistent results over short distances.

Comparison of results for all biological methods, independent of location, shows a consistent correlation (Fig. 8). No single

method stands out as producing persistently divergent results from other methods, although all biological methods produce uplift estimates that are higher than LiDAR results. The tide gauge calibrated method has yielded results most consistent to LiDAR. At all sites uplift estimated using the tide gauge calibration method give results within 0.0 to +0.21 m (or 0.35 to 21%) higher than LiDAR results, with a mean of +0.11 m (10%). Further, at all sites and over all biological methods, uplifts estimates are 0.0 to +0.31 m (or < 34%) higher than associated LiDAR results, with a mean of + 0.17 m.

## 5 Discussion


The distribution of kelp within the intertidal zone at Kaikōura is well defined with respect to qualitative zones of upper, mid and low intertidal (Marsden, 1985). Nevertheless, because the width of the intertidal zone varies with site exposure, topography, wave-wash and competition between different organisms, an attempt to quantify this uncertainty is made by





calibrating coseismically uplifted intertidal brown algae (*Durvillaea* and *Carpophyllum*) in the immediate vicinity of the
Kaikōura Tide Gauge, aiming to establish a quantitative correction value for the living-depth of the kelp holdfasts with
respect to MLWS (Figs. 3 and 5).

Using Eq. (1) (see Sect. 3.2) at the Kaikōura Tide Gauge, a living depth correction factor $X_C$ of $0.26 \pm 0.09$ m above MLWS
is derived for sheltered *Carpophyllum maschalocarpum*. For *Durvillaea* in sheltered sites, a living depth correction factor $X_D$
of $0.38 \pm 0.09$ m above MLWS is derived. These values were subsequently used to estimate tectonic uplift at sites located up
to 15 km from the tide-gauge and produced uplift measurements which were in good agreement with uplift calculated at the
same localities by differential LiDAR (Figs. 6 and 7). Thus, it appears that this method of estimating correction values may
be important as it provides, for the first time, an independent quantitative method for estimating the preferred water-depth of
intertidal biota with respect to MLWS. This method may be applied elsewhere to other intertidal biota in the vicinity of a
tide-gauge. *Carpophyllum* is endemic to New Zealand while *Durvillaea* is widespread in the southern hemisphere. The
derived correction values are specific to these taxa at Kaikōura region which is characterised by a moderate tidal range. If
these values are applied elsewhere, the uncertainty would equal to the maximum correction value of 0.38 m. The three
biological post-processing methods used to obtain uplift, they all yield results which are, within uncertainties, similar to one
another, meaning that any of these methods could be applied depending on the available tidal data at the site of interest.
Analysis of all data suggest that hand-held measurements most often overestimate uplift, with results higher for tape-
measure data than RTK-GPS survey measurement.

In the vicinity of the Kaikōura Tide Gauge, biological results using the tide-gauge correction factor are the most similar to
non-biological methods. With increasing distance from the tide-gauge, this new method provides reliable results;
nevertheless, other biological methods were comparable. Progression of daily tides is even and fluctuations from the
expected tidal progression may occur over several minute intervals due to natural unevenness in the ocean surface caused by
wind, barometric pressure and local topography (eg. Garrison, 2010). While the influence of this natural fluctuation for
biological data collected proximal to the tide-gauge is well mitigated by use of real-time tide gauge water-level (H), away
from the Kaikōura Tide Gauge this real time fluctuation is less able to be mitigated. Therefore the NIWA and LINZ tidal
chart calculations for H, and associated correction factors may give equally accurate uplift estimates. Overall, the NIWA
method produces results more consistent with non-biological methods than does the LINZ method. Despite this, data
collected by RTK and processed using predictive charts, such as LINZ, may be used to calculate uplift estimates, and could
be used with confidence in remote locations, or locations where other methods are not available.

This study has shown that instrumental and biological methods can produce comparable results; yet, in order to reduce
uncertainty in the biological methods, the biota should have a living-depth relative to an appropriate sea-level datum that is
calibrated against real-time tide-gauge data. Towards this direction, our study has provided a new calibration method to



derive a correction factor for this living-depth that can be applied globally where tide-gauge records are available. In circumstances where tide-gauge records are unavailable, the usage of predictive charts to process biological data may still be appropriate, accepting that uncertainties may be higher.

## 6 Conclusions

Tectonic deformation determined from uplifted intertidal biozone indicators produce results comparable with tectonic uplift recorded by the Kaikōura tide-gauge, remote-sense datasets (LiDAR and RTK-GPS) and strong-motion seismic data. Calibrating measured intertidal biological data to real-time tide gauge records gives results within an average 0.11 m of those derived from direct uplift of the tide-gauge, and localised differential LiDAR values. Uplift results from biological data, calibrated using predictive tidal charts, are as reliable as other biological and non-biological methods when distant to real-time tide-gauges, and are appropriate for use where differential LiDAR or other real-time remote-sense datasets are not available. Results from this study indicate that *Carpophyllum,* an alga with a tightly defined upper intertidal limit, is the most reliable predictor of uplift at sheltered sites. *Durvillea*, an alga with a less well-defined upper intertidal limit, is less reliable, especially when measured at exposed sites. Biological data collected by RTK-GNSS gives the strongest overall comparison to non-biological methods of estimating uplift. Data collected by tape-measure may be reliable where sheltered sites are available, but are likely to give higher apparent uplift results in exposed locations, where intertidal biozones are blurred and elevated by wave fetch and exposure on sections of a rocky coastline.

**Author's contributions**
All authors contributed to the research idea, data-collection during fieldwork and their subsequent analysis. The first draft of the manuscript was written by C.R. with the contribution of all co-authors. The final manuscript resulted from close collaboration of all co-authors.

**Acknowledgements**
This work was partly funded by a HART-GFZ grant. Thanks to Kate Clark and colleagues at GNS Science (Lower Hutt) and Sharyn Goldstien and Islay Marsden (Biological Sciences, University of Canterbury) for early discussion on methodology and distribution of intertidal biota at Kaikōura. Thanks also to Rongjang Wang (GFZ) for providing his *smbloc*-code for the calculation of static offset from strong-motion data and to Dick Beetham for his able assistance during fieldwork.

**Competing interests**
The authors declare that they have no conflict of interest





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



**Figure and table captions**

**Figure 1:** (a) Inset map of New Zealand illustrating the main tectonic features of the Hikurangi subduction margin, the location of the Marlborough Fault System (MFS) and the epicentre of the 2016 $M_w$ 7.8 Kaikōura Earthquake. Blue box near the Kaikōura Earthquake epicentre indicates the study area. (b) Map showing the study localities from which *Durvillaea* and
*Carpophyllum* holdfast measurements were recorded with RTK GNSS, the position of State Highway One (SH1) from which LiDAR data points were derived, the location of Kaikōura Tide Gauge and the KIKS strong ground motion station. The Hundalee Fault is also illustrated. Background image supplied by Land Information New Zealand.

**Figure 2:** Field photographs of the intertidal zone and biota near Kaikōura (taken after the earthquake). (a) Healthy
*Durvillaea* (mostly *D. willana*) (D) and *Carpophyllum* (C) photographed at low-tide.  (b) uplifted bedrock north of Paia Point showing living *Carpophyllum* (C) and dead *Carpophyllum* holdfast stumps (CH). Note also the living pink coralline algae at the waterline and bleached morbid coralline algae (arrows). (c) uplifted intertidal zone near the Kaikōura tide gauge, showing distinctive line of *Carpophyllum* holdfasts (CH) and dispersed *Durvillaea* holdfasts (DH). (d) uplifted intertidal zone near Paia Point. One of the authors (J.B.) measures the elevation of *Durvillaea* holdfasts (DH) and *Carpophyllum*
holdfasts (CH) using RTK GNSS survey equipment. Note distinctive white zone of dead coralline algae.

**Figure 3:** Schematic diagram illustrating uplift and stranding of holdfasts at the Kaikōura Tide Gauge. It also illustrates schematically the method for calculating the offset of holdfasts ($X_{C/D/G}$; Eq. 1) from Mean Low-Water Spring (MLWS) of the tide-gauge data. MLWN = Mean Low-Water Neap, MHWN = Mean High-Water Neap, MHWS = Mean High-Water
Spring, $H_{TG}$ = tide height as measured in tide gauge, $U_{TG}$ = uplift as measured by tide gauge offset data pre- and post-deformation, OM = observed measurement (holdfast), X = offset of holdfasts from MLWS. Inset: Results for X as calculated for kelp at the Kaikoura Tide Gauge. Mean values are shown by a solid-circle while tails represent maxima and minima values. See Sect. 3.2.1 for details.

**Figure 4:** Schematic diagram illustrating uplift and stranding of holdfasts used to calculate offset of holdfasts ($X_{NIWA/LINZ}$) from Mean Low-Water Spring (MLWS) independent from tide gauge data. MLWN = Mean Low-Water Neap. Here MLWS is determined from LINZ and NIWA predictive charts, and the position of holdfasts with respect to MLWS and MLWN is determined from local knowledge of kelp distribution (Goldstien pers. comm. 2017).

**Figure 5:** Uplift measured at the Kaikoura Tide Gauge from various kelp holdfasts and exposure sites plotted against the offset recorded, at the same locality, by the Tide Gauge and differential LiDAR. Holdfast data are presented as *mean* and *standard deviation* while the Tide Gauge and LiDAR data are presented as *mean* only.



**Figure 6:** (a) Tectonic uplift in metres measured at the Kaikōura Tide Gauge, Kaikōura Harbour, Paia Point and Omihi Point
from biological data processed using the tide-gauge correction, and NIWA and LINZ predictive tide-chart correction
methods (see Sect. 3.2). These values are compared to uplift recorded by the Tide Gauge and differential LiDAR (where
available). (b) Percentage of uplift-deviation of the biological methods with respect to the LiDAR measurements. Horizontal
axis not to scale.

**Figure 7:** Locality, digital elevation imagery and differential LiDAR data for Paia Point (see Fig. 1 for location). (a) Aerial
photo from ©Google Earth imagery of Paia Point, State Highway-1 and uplift collection points. Blue line = portion of SH1
from which differential LiDAR uplift calculated, red circles: RTK-GPS collected kelp data-points, yellow circles: tape-
measure collected kelp data-points. (b) Digital Elevation Model developed from post-earthquake LiDAR data. Blue line and
colour-coded circles as per (a). (c) Plot of uplift of points at 1 m intervals along the blue line on SH1 in (a) and (b) derived
from differential LiDAR.

**Figure 8:** Cross plots of data collection and processing methods. (a) RTK and tape-measure uplift data processed using the
tide-gauge correction method. (b) RTK uplift data, processed using the tide-gauge correction method, plotted against
differential LiDAR uplift data. (c) Tape-measure uplift data, processed using tide-gauge correction method, plotted against
differential LiDAR uplift data. (d) RTK uplift data, processed using the tide-gauge correction method, plotted against the
NIWA Forecaster tide chart correction method.

**Table 1:** Calculation of uplift at the Kaikōura Tide Gauge (KAIT) using tide-gauge readings for high and low spring-tides
and high and low neap-tides (see Sect. 3.1).

**Table 2:** Absolute uplift values calculated from the Kaikōura Tide Gauge data using methods B-E. Method B: Comparison
of average high-tide and low-tide readings from several tidal cycles (3-day period) before and after the earthquake; Method
C: Aligning pre-earthquake tidal data with post-earthquake data and incrementally adjusting them until a best fit; Method D:
comparing the average water-elevation from a pre-earthquake month to the same month's data after the earthquake
(December 2015 against December 2016); Method E: Calculating the difference in average waterline elevations for an
extended period (44 days) before and after the earthquake (Nov 14[th] to Dec 27[th]). As Method A here we refer the
methodology established in Sect. 3.1 and presented in Table 1.

**Table 3:** Results for calculation of the living-depth $X_{C/D/G}$ relative to MLWS for holdfasts at the Kaikōura Tide Gauge. Note,
that only holdfasts of *Carpophyllum* and *Durvillaea* in sheltered locations were used to calculate this depth.



**Table 4:** Comparison of uplift results for data collected by RTK and tape-measure at the tide-gauge, and including a comparison of kelp types in both sheltered and exposed locations. Results are presented by holdfast species and exposure ranking, independently of the collection site.


**Table 5:** Comparison of mean uplift values derived using RTK for the various methodologies (e.g., Tide Gage calibration method, NIWA Forecaster method, LINZ Tide Chart method). As the source data remain the identical for each method, the standard deviation reflects error derived from the RTK measurements. Data is presented by site at each location; where a site was collected using both *Carpophyllum* and *Durvillaea,* the holdfast type is recorded as "mixed".


**Table 6:** Uplift calculated from differential LiDAR and strong-motion uplift estimated from the KIKS station.





**Figure 1**






**Figure 2**





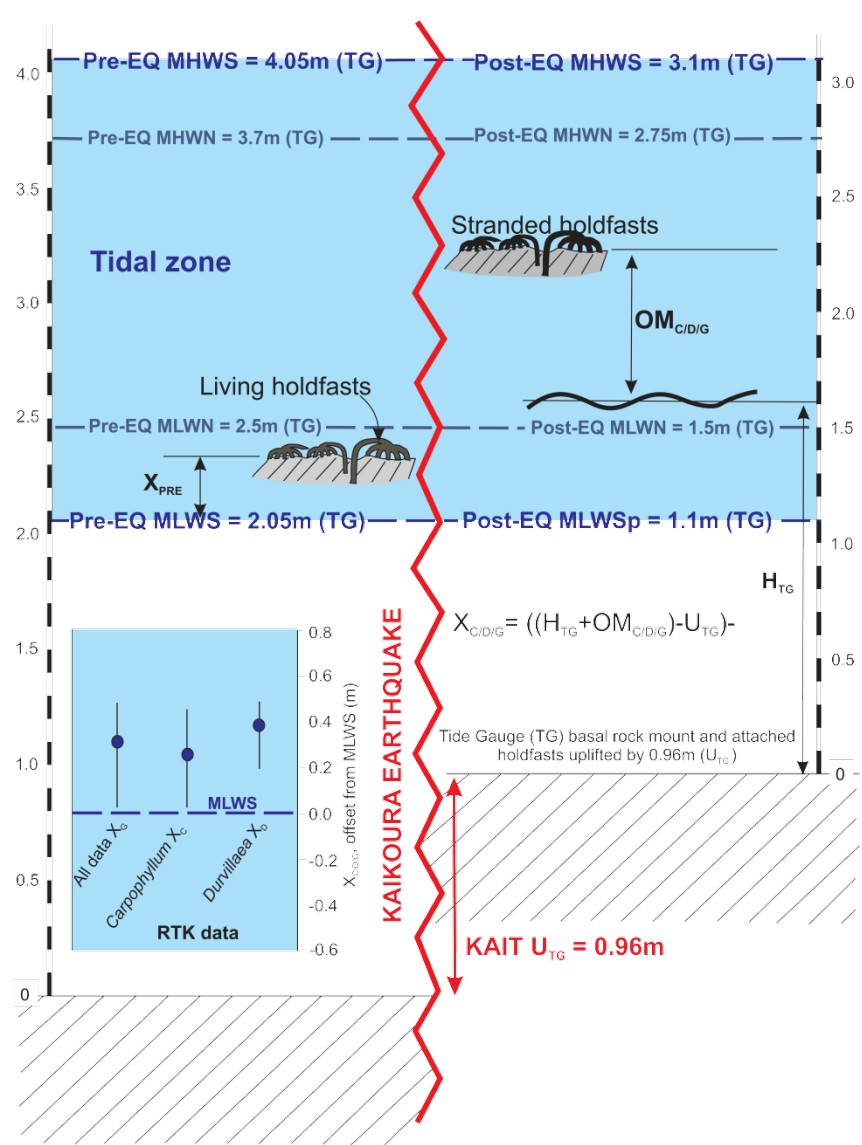


**Figure 3**


Earth **Surface**
**Dynamics**
Discussions

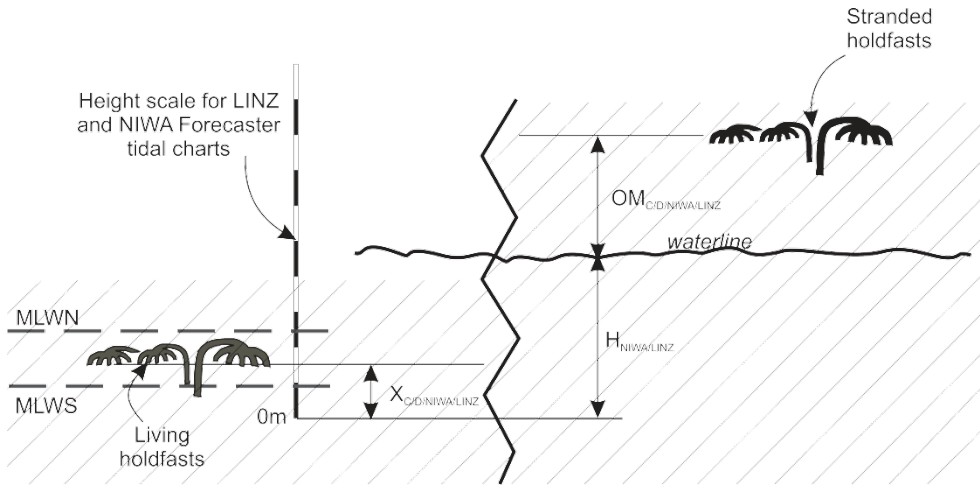


**Figure 4**

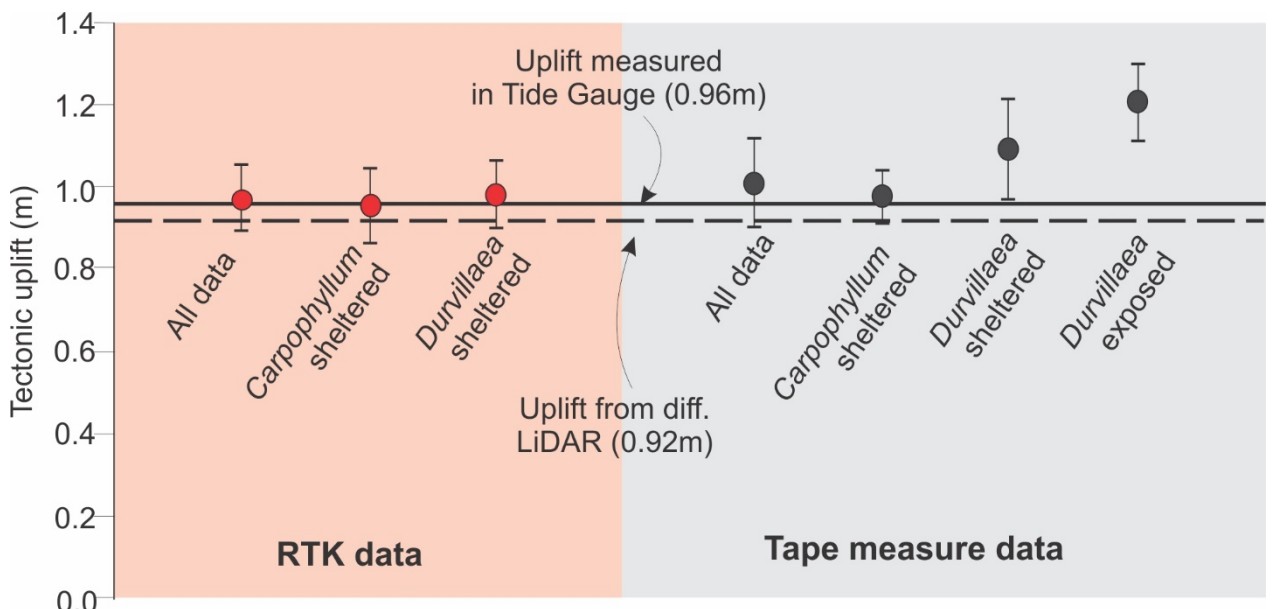


**Figure 5**






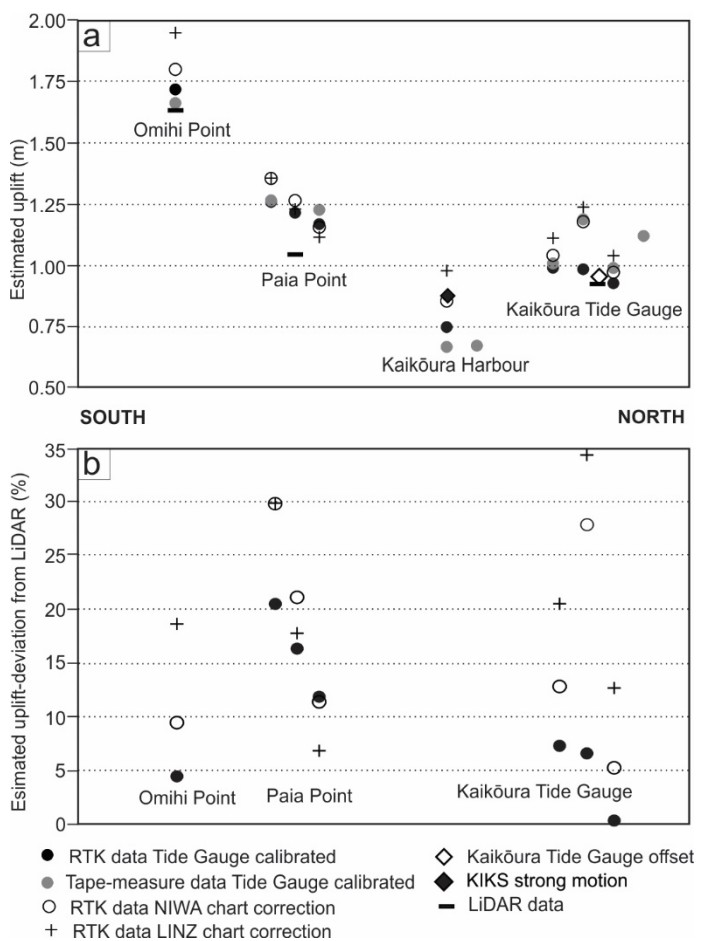

**Figure 6**





**Figure 7**







**Figure 8**




|  | Spring tide | | Neap tide | | Uplift | |
|---|---|---|---|---|---|---|
|  | Pre-EQ | Post-EQ | Pre-EQ | Post-EQ | Spring diff | Neap diff |
| **High tide** | 4.05m | 3.1m | 3.7m | 2.75m | 0.95m | 0.95m |
| **Low tide** | 2.05m | 1.1m | 2.5m | 1.5m | 0.95m | 1m |
| **Range** | 2m | 2m | 1.2m | 1.25m | **Mean diff** | **0.96m** |


**Table 1**

| Method | Data points | Mean uplift (m) |
|---|---|---|
| A |  | 0.96 |
| B | 6 | 0.95 |
| C | 17568 | 0.98 |
| D | 44640 | 0.96 |
| E | 17932 | 0.97 |
| **Overall mean uplift $U_{TG}$** | | **0.96** |
| **Standard deviation** | | **0.02** |

**Table 2**







|  | **Mean** | **SD** | **Median** | **Max** | **Min** |
|---|---|---|---|---|---|
| **All holdfasts $X_G$** | 0.31m | 0.10m | 0.32m | 0.50m | 0.01m |
| ***Carpophyllum* $X_C$** | 0.26m | 0.09m | 0.26m | 0.43m | 0.01m |
| ***Durvillaea* $X_D$** | 0.38m | 0.07m | 0.39m | 0.50m | 0.19m |

**Table 3**


|  | **Mean (m)** | **SD (m)** | **Min (m)** | **Max (m)** |
|---|---|---|---|---|
| **RTK data** |  |  |  |  |
| All data | 0.97 | 0.08 | 0.71 | 1.13 |
| *Carpophyllum* sheltered | 0.96 | 0.09 | 0.71 | 1.13 |
| *Durvillaea* sheltered | 0.98 | 0.07 | 0.78 | 1.09 |
| **Tape measure data** |  |  |  |  |
| All data | 1.05 | 0.11 | 0.87 | 1.35 |
| *Carpophyllum* sheltered | 0.98 | 0.06 | 0.87 | 1.13 |
| *Carpophyllum* exposed | 1.06 | 0.07 | 0.92 | 1.22 |
| *Durvillaea* sheltered | 1.10 | 0.13 | 0.91 | 1.35 |
| *Durvillaea* exposed | 1.21 | 0.09 | 1.07 | 1.35 |

**Table 4**







| Site | Holdfast type | Tide Gauge Mean (m) | NIWA Forecaster Mean (m) | LINZ Tide Charts Mean (m) | SD* (m) |
|---|---|---|---|---|---|
| **RTK** | | | | | |
| Tide Gauge 1 | *Carpophyllum* | 0.99 | 1.04 | 1.11 | 0.06 |
| Tide Gauge 2 | *Carpophyllum* | 0.92 | 0.97 | 1.04 | 0.10 |
| Tide Gauge 3 | *Durvillaea* | 0.98 | 1.18 | 1.24 | 0.07 |
| Paia Point 1 | *Carpophyllum* | 1.27 | 1.27 | 1.24 | 0.11 |
| Paia Point 2 | *Durvillaea* | 1.22 | 1.36 | 1.36 | 0.18 |
| Paia Point 3 | Mixed | 1.18 | 1.17 | 1.12 | 0.16 |
| Omihi Point 1 | *Carpophyllum* | 1.71 | 1.80 | 1.95 | 0.13 |
| Kaikōura Hbr | *Carpophyllum* | 0.74 | 0.85 | 0.98 | 0.12 |
| **Tape Measure** | | | | | |
| Tide Gauge 1 | *Carpophyllum* | 1.00 | | | 0.07 |
| Tide Gauge 2 | *Carpophyllum* | 1.12 | | | 0.11 |
| Tide Gauge 3 | *Durvillaea* | 1.19 | | | 0.08 |
| Tide Gauge 4 | Mixed | 0.99 | | | 0.06 |
| Paia Point 1 | Mixed | 1.27 | | | 0.09 |
| Paia Point 2 | Mixed | 1.23 | | | 0.09 |
| Omihi Point 1 | Mixed | 1.66 | | | 0.17 |
| Kaikōura Hbr | *Carpophyllum* | 0.66 | | | 0.10 |
| Kaikōura Hbr | *Carpophyllum* | 0.67 | | | 0.06 |


**Table 5**







| | Mean (m) | Median (m) | SD (m) | Max (m) | Min (m) |
|---|---|---|---|---|---|
| **Differential LiDAR** | | | | | |
| Tide Gauge | 0.92 | 0.91 | 0.06 | 1.13 | 0.77 |
| Paia Point | 1.05 | 1.05 | 0.07 | 1.31 | 0.90 |
| Omihi Point | 1.64 | 1.64 | 0.04 | 1.74 | 1.53 |
| **KIKS Strong motion** | | | | | |
| Kaikōura Harbour | 0.87 | 0.06 | | | |

**Table 6**