# Peer review of "Using a calibrated upper living-position of marine-biota to calculate coseismic uplift: the case-study of 2016 Kaikōura Earthquake, New Zealand"

_Earth Surface Dynamics, 2019_

## Referee Comment (RC1) · Brendan Duffy (Referee) · 9 Feb 2020

School of Earth Sciences
McCoy Building
The University of Melbourne,
Victoria 3010, Australia

9 Feb. 20

Dear Editor,

Re: Reid et al. : A new method for calibrating marine biota living-depth using the 2016 Kaikōura Earthquake uplift

Thank you for the opportunity to comment on this manuscript. The authors have developed a method of using tide tables and ecological understanding of tidal biozones to determine coseismic surface uplift from the elevation of stranded algal anchors. They present these results alongside biological estimates underpinned by a tide gauge calibration, and compare the results of all with the vertical component of differential lidar and locally a strong motion sensor. Their short discussion summarizes the results and discusses the systematic deviation from the Lidar measurements.

The method is novel and but the paper unfortunately lacks punch in the discussion, which goes nowhere really. The first two paragraphs are basically a summary and the third paragraph discussion of systematic deviation from Lidar measurements of uplift (figure 5b) and what they mean for the calibration is limited to invoking fluctuations. If the fluctuations are on the order of minutes, and the tide level was measured repeatedly at a site, can the consistent high estimates be put down to fluctuations? At line 427 (While the influence…) they say that fluctuations in the tide are mitigated by using tide gauge, but surely the estimates there were obtained using the various correction factors and local sea level measurements. My reading is that the tide gauge was only used to establish the correction, and not thereafter.

Intuitively, one would expect the RTK-tide gauge correction to be the most robust. That is borne out by the RTK-tide gauge correction around the tide gauge, which was within plus/minus 7 cm of correct while tide table corrected results varied more widely. At other sites (e.g. Paia point) each group of individual assessments are within 5% of each other but there can be 20% discrepancy from one group to the next, and with respect to the Lidar estimate. It seems to me as though the correction is reasonably precise (suggesting that the underlying concept is robust) but also quite inaccurate.

If I was a coastal ecologist I would be interested in that. Obviously local wave climate or tidal fluctuations can be a factor, as they discuss, but it may also be more interesting than that. The X-factor is a positive elevation value and is subtracted, so if the uplift is too high (most places), then not enough has been subtracted and the organism is actually shallower dwelling relative to MLWS. If the uplift value is too low, then too much has been subtracted and the organism is deeper dwelling relative to MLWS (Kaikoura harbour). Both the tide gauge and the harbour are presumably very sheltered and also areas of boat traffic, which must have some impact on marine algae distribution. The only site that yielded a too-low estimate was Kaikoura Harbour, where boat traffic and maybe sheltering is greatest. Given that in most cases the correction is too small, it may be that the organisms actually range further above MLWS than expected, while still remaining below MLWN. At Paia Point, it seems that the algae attached to rocks furthest to seaward are the most undercorrected, suggesting that those rocks (in an area where they will be bathed regularly by swell and wave action, even at the lowest tides) have the shallowest depth range.

Another thing that would be informative is to know what timescale this technique is available over. The authors carried out their work after a few months, but how much longer could they have realistically applied the technique. Also, is the 20 holdfasts that seems to be their lower limit statistically valid. If the authors take their first five, ten, twenty measurements at a site, does the result change significantly. Would it improve if they used fifty? Maybe start by showing the number of measurements of each type on Figure 5. Maybe somehow on Figure 6 too.

One strategy for the discussion is already present in the introduction. The authors provide an extensive list of coastal uplifts and biological assessments thereof in the introduction. I have always thought that the discussion should revisit the key points of the introduction. So, please revisit that list and discuss the advantages and limitations of this technique. How many historic earthquakes have caused coastal uplift, what magnitudes of coastal uplift have been documented, using what biological indicators. Which of them could have been targeted with this technique, and over what timescale. Maybe put together a useful chart/table showing the preservation potential, accuracy, precision, ease of deployment, best vertical resolution (critical if you want to document uplifts of half a meter or less), speed of survey, skill requirements, etc of the various techniques and illustrating why this one is important – I am thinking something like Table 2B in Quigley et al. (2016), in which the lead author was responsible for another biological assessment of vertical displacement. Or maybe a McCalpin style graphic. Whatever you do, please re-read your introduction and use it to put some spice into the discussion.

These are just a few ideas, but I really think the author team is uniquely qualified to lead this discussion towards a useful earthquake-ecology viewpoint, especially given the lead author's established expertise in biological zonation (Reid et al., 2011 - their fig 2 deserves a citation here) and earthquake effects on that in New Zealand (Quigley et al., 2016). One way or another the discussion needs to be beefed up. A brief discussion of some of these points, possibly using these results to think further about some previous Reid et al work, would probably find favour with coastal ecologists and increase the citability of the paper. A table or graphic of the kind I suggest above would surely contribute to the quantitative coastal ecological impact assessments that will follow future earthquakes, in New Zealand and elsewhere.

Enough long-winded discussion of the discussion (eek). Another key criticism relates to the description of the methods. I found this opaque, with many ambiguities. The opacity is largely because the paper dives straight into formulae without really explaining the strategy. This is not helped by a major discrepancy between the formula shown in text and that shown in Figure 3. After some careful checking I am happy that the methods are valid but they need to be clarified and subscripts used consistently. I also see little indication of the uncertainties in the figures. They are covered to a certain extent in the text, but there are no error bars on major figures (e.g. Fig 6).

Below I provide detailed comments and corrections, mostly regarding the methods and description thereof. There may be some errors of understanding on my part and I apologise for any such errors contained here.

Once again, thank you for this opportunity and good luck to the authors. I look forward to seeing this published.

Sincerely,

Brendan

| LINE | COMMENT |
|---|---|
| 25 | Satellite data is ubiquitous, Lidar replacement is the real target here. |
| 31 | Surely Darwin could get a mention here? |
| 78 | Not convinced this is the right reference. What about Williams et al. (2013)? |
| 83 | An earthquake network comprises several faults. An earthquake is a process that ruptures a fault or a network of faults. An earthquake does not comprise a network of faults. |
| 97 | Vaguely and unintentionally implies that the mapped surface faults and coastal uplift, as well as the tsunami, extended 250 km south of Kaikoura. |
| 102 | Here and elsewhere (e.g., line 103, 105, 114…), the word 'exposed' and derivatives of that word are used in two different senses of the word – Exposed coastlines and exposed holdfasts. Sometimes it is clear from context and sometimes it requires a double read to figure it out. Please consider using expos… in one sense only and replacing the other meaning with a different word. Line 114 is particularly bad – controlled by exposure above the tide? Lack of shelter? It is not an easy issue to address but maybe keep exposed for 'unsheltered' and use qualifications such as subaerially-exposed, terrestrially-exposed, outcropping, etc for stuff that is above sea level. |
| 106 | It is not diurnal, it is semi-diurnal, with two full cycles daily. |
| 188 | It seems to me that you never explicitly state the tide gauge uplift except in Table 1. Why not? |
| 189 | Change to "Biological data collection" and then add a new section title at line 220 - "Data processing" |
| 211 | Were the wave effects given a plus-minus value? |
| 213 | Was sea-level remeasured after each group of twenty? |
| 215 | 215-218 – As somebody with building experience I would have forgotten about the tape and used either 1) a builder's laser level and a reflective staff. Measure the height of the laser level mount with a tape, then measure the height of holdfasts in all positions using a reflective staff. Laser levels are small, portable, and cheap as chips and the staff could be a stick with a high vis jacket. Any holdfast accessible with RTK could be done with a laser level, especially in the late evening, and would yield similar accuracy to RTK. 2) Even cheaper, a homemade water level, with the reservoir placed on a local high point. Engineers used commercial versions widely after the Chch earthquake to survey floor levels. (https://en.wikipedia.org/wiki/Water_level_(device)) |
| 220 | New section title here - maybe "3.3 Data Processing". |
| 221 | These field measurements of apparent uplift… [No. they are field measurements of exposure above a reference tide level. Nobody would consider that to be apparent uplift because it is a time-dependent measurement] were then further processed to determine the total uplift [No. uplift is either of rock or surface – in this case both are equivalent at this moment in time, so just say surface uplift], taking into account the time of data measurement and the pre-earthquake living position of the algal holdfasts which is the difference between pre- and post-earthquake elevation of algal holdfasts [Note that position is a 3D thing and we are only interested in z, not x or y]. |
| 222 | Just say "Three different corrections were used to derive surface uplift from elevation above sea level at a point in time. These were a) tide gauge calibration; b) interpolation of NIWA tide forecaster and c) interpolation of LINZ tide forecaster. Method a) calculated a correction using direct measurement of stranded algae relative to the |

| | |
|---|---|
| | known uplift of the Kaikoura tide gauge, whereas b) and c) combined the tide forecaster interpolation with local knowledge of biological living zones. Although more subjective, corrections b and c are applicable to sites where no tide gauge is available. |
| 229 | A factor is a number that is multiplied by another to yield a product. This is not a factor, it is a constant. There is also more than one constant. |
| 229 | I suggest that you change title to "Correction a) Deriving living depth constants for target species using the Kaikoura tide gauge." |
| 230 | 230-250 – Try and be a little kinder to those who want to reproduce this. I found it very hard to follow until I figured out what you were doing. Maybe describe in words what you are doing, rather than diving straight into the derivation of your constant. I recommend: "Surface uplift is the difference between the present, post-uplift elevation and the pre-uplift elevation (the living elevation) of the organisms holdfasts. The target species occupy slightly different living positions in the inter-tidal zone, so this new method first derives a constant living elevation (X, in m) for each target species (XC – Carpophyllum; XD – Durvillaea; Combined XG), by calculating the pre-earthquake elevation of the stranded holdfasts relative to the spring-tide mean low water level (MLWS). The constant XC/D/G is calculated in three stages, using holdfasts at sheltered sites close to the Kaikoura tide gauge. First the height of the stranded holdfast above the uplifted tide gauge mount is calculated from the sum of the tide gauge height and the observed elevation of the stranded holdfast relative to sea level (both at measurement time). Secondly, the tide gauge uplift is subtracted obtain the elevation relative to the pre-quake tide gauge. Finally, MLWS is further subtracted to obtain elevation relative to MLWS, which is a key reference level for the biological zonation. This procedure is given by the equation: ... where |
| 231 | Depth implies below sea level. At least one species lived above $MLW_S$, so depth would be negative. Just use elevation, since you are differencing with elevation. |
| 235 | Bit of a mess here. Outer parentheses are redundant. The MLWS subtraction is not shown in the equation on Figure 3. Figure 3 uses $H_{TG}$ instead of H, so please be consistent. |
| 240 | subscripts rather than indices? |
| 241 | … average **post-uplift** tide gauge reading… |
| 244 | You have already said they occupy different levels, especially if you adopt the text above, so just leave out the first sentence of this paragraph. |
| 247 | "Further, only …." – This should appear within the derivation of the formula. See comment for line 230 above. |
| 256 | You introduce a parameter without defining it (UB(TG)). Sure I can work out what it is but I shouldn't need to. Also H in eqn 2 looks a lot like H in eqn 1. Hence the need to use HTG in eqn 1. Each time you use H it refers to a specific time and place. I wonder if you need to have HTG and HSS (survey site). One way or another your subscripts need to be unique, informative and consistent, because at the moment they are not. I recommend that you include a glossary of the terms used in your equations |
| 265 | Please clarify this process. Maybe change text beginning line 265 to read... "where OMCDG is the observed elevation of the holdfasts relative to locally measured sea level, HNIWA/LINZ is the difference between the predicted tide height at survey time and the MLWS based on one year of predictive tables, and XC/D relies on expert assessment as follows: Carpophyllum..." Note here, please be consistent with subscripts. I cant see any reason why OM here would be different to OMC/D/G in equation 1. Also, how would you get an expert assessment of tidal zonation in Timor for instance? |

| | |
|---|---|
| **272** | What is a regional height |
| **274** | Replace height with holdfast elevation (height is a vague term) |
| **285** | Why are NIWA and LINZ different? Can you illustrate the difference in a figure, or summarize it somehow? |
| **292** | The absolute accuracy of RTK may be 5 cm vertical if you put your base station on a suitable order trig. The internal relative accuracy is better than that (2 cm) with favourable GPS environment (see e.g table 1 in Duffy et al. 2013). On that note, where was your base station set up? Maybe it doesn't matter, if all of your RTK measurements were differenced with a local sea level measurement every 10 minutes or so but you should still mention it. Personally, I would have opened and closed the survey at the tide gauge, so that I could see how well my sea level measurements replicated the tide gauge. |
| **314** | Please check figure order. I haven't checked super carefully, but Fig 7 seems to come before 5 and 6. Please make sure they are numbered in order. |
| **320** | The problem with the lidar at Kaikoura harbour should be mentioned here, not in the results. And really, I want to know how inconsistent they were. If you do a histogram over a couple of roads around the harbour, what do you get? Is it consistent with the strong motion instrument or not? |
| **336** | New heading - 4.1 Tide gauge locality |
| **339** | Delete sentence - "Uplift estimates derived…" |
| **344** | … tide gauge (Figure 5) and compared with uplift of the Kaikoura tide gauge (calculated in section 3.1) |
| **370** | New section title here - maybe "4.2 Paia Point and Omihi Point". |
| **392** | How inconsistent? And why? Differential beach gravel compaction? Something else? Does this inconsistentcy affect have any effect over the distance from strong motion sensor to measurement sites? |
| **432** | Basically repeats something you have said in 360 and in 226. Just say it once, with maximum impact. |
| **FIGURE 2** | In the caption, mention green ulva (from line 120) |
| **FIGURE 3** | Fix the missing bit of the formula. Shift the Rtk data panel somewhere else or at least change the background colour, because as drawn it looks like part of the conceptual panel. |
| **FIGURE 4** | Not really correct. Why is an equation shown on Figure 3 but not on Figure 4? Surely $X_{C/D\_NIWA\_LINZ}$ is measured from MLWS?? |
| **FIGURE 6** | Caption - note that no lidar comparison was produced for Kaikoura harbour. Even better, show the real picture with appropriate error bars. |
| **FIGURE 7** | part b - No Rainbow, not now, not ever. Use a proper colour stretch, and stretch it over the elevation range from sea to just above the road. I don't particularly care about the hill. |

Quigley, M. C., Hughes, M. W., Bradley, B. A., Ballegooy, S. v., Reid, C., Morgenroth, J., Horton, T., Duffy, B., and Pettinga, J. R.: The 2010-2012 Canterbury earthquake sequence: Environmental effects, seismic triggering thresholds and geologic legacy, Tectonophysics, 672-673, 228-274, 2016.
Reid, C. M., James, N. P., and Bone, Y.: Carbonate sediments in a cool-water macroalgal environment, Kaikoura, New Zealand, Sedimentology, 58, 1935-1952, 2011.

---

## Referee Comment (RC2) · Anonymous Referee #2 · 21 Feb 2020

Dear editor, Thank you for giving the opportunity to review the paper by Reid et al. I read with interest the submitted manuscript and overall I believe it is an interesting contribution fitting the scope of the journal. The authors use tidal data and the ecological preference of intertidal algae to determine coastal uplift. They calibrate their biological data with both real-time and predictive tidal charts and compare their results with lidar and strong motion data. Overall, I think their proposed methodology will be of interest to many coastal geomorphologists and geologists working with coastal deformation and their paper deserves publication. However, in order for their methodology to be replicated some revisions are necessary, primarily in the methods section. In particular, paragraphs 3.2.1 – 3.2.3 are quite hard for a reader to follow and more complicated

that necessary. I suggest the authors to improve this part by explaining in a better way their methodological approach. Some technical corrections line 376: include the error bar line 389-390: error bar for NIWA and LINZ? line 418: middle on line, "they" is not needed Caption of figure 1: line 570, note that the position of State 570 Highway One is the yellow line Figure 6: I would suggest to include error bars

---

## Author Comment (AC1) · 19 Mar 2020

Response to Brendan Duffy

Important note: our responses to Brendan Duffy's comments are in red italics (in the attached pdf) and the line numbers in our responses refer to the revised version of the manuscript.

Dear Editor,

Re: Reid et al. : A new method for calibrating marine biota living-depth us i ng the 2016 Kai kÅ■u ra Earthquake uplift

[Figure]

Thank you for the opportunity to comment on this manuscript. The authors have developed a method of using tide tables and ecological understanding of tidal biozones to determine coseismic surface uplift from the elevation of stranded algal anchors. They present these results alongside biological estimates underpinned by a tide gauge calibration, and compare the results of all with the vertical component of differential lidar and locally a strong motion sensor. Their short discussion summarizes the results and discusses the systematic deviation from the Lidar measurements.

The method is novel and but the paper unfortunately lacks punch in the discussion, which goes nowhere really. The first two paragraphs are basically a summary and the third paragraph discussion of systematic deviation from Lidar measurements of uplift (figure 5b) and what they mean for the calibration is limited to invoking fluctuations. If the fluctuations are on the order of minutes, and the tide level was measured repeatedly at a site, can the consistent high estimates be put down to fluctuations? At line 427 (While the influence. . .) they say that fluctuations in the tide are mitigated by using tide gauge, but surely the estimates there were obtained using the various correction factors and local sea level measurements. My reading is that the tide gauge was only used to establish the correction, and not thereafter.

Intuitively, one would expect the RTK-tide gauge correction to be the most robust. That is borne out by the RTK-tide gauge correction around the tide gauge, which was within plus/minus 7 cm of correct while tide table corrected results varied more widely. At other sites (e.g. Paia point) each group of individual assessments are within 5% of each other but there can be 20% discrepancy from one group to the next, and with respect to the Lidar estimate. It seems to me as though the correction is reasonably precise (suggesting that the underlying concept is robust) but also quite inaccurate.

If I was a coastal ecologist I would be interested in that. Obviously local wave climate or tidal fluctuations can be a factor, as they discuss, but it may also be more interesting than that. The X- factor is a positive elevation value and is subtracted, so if the uplift is too high (most places), then not enough has been subtracted and the organism is

actually shallower dwelling relative to MLWS. If the uplift value is too low, then too much has been subtracted and the organism is deeper dwelling relative to MLWS (Kaikoura harbour). Both the tide gauge and the harbour are presumably very sheltered and also areas of boat traffic, which must have some impact on marine algae distribution. The only site that yielded a too-low estimate was Kaikoura Harbour, where boat traffic and maybe sheltering is greatest. Given that in most cases the correction is too small, it may be that the organisms actually range further above MLWS than expected, while still remaining below MLWN. At Paia Point, it seems that the algae attached to rocks furthest to seaward are the most undercorrected, suggesting that those rocks (in an area where they will be bathed regularly by swell and wave action, even at the lowest tides) have the shallowest depth range.

Response – yes, wave action is a factor in the distribution of inter-tidal species. The higher the wave exposure the higher the wave splash and, therefore, the higher the extent of species. However species respond over a seasonal or average annual cycle, and thus at the time of visiting it may be impossible for the geologist viewer to measure this effect, and it is inherent as potential error in our methodology. As described by the reviewer this effect has become apparent in a semi-measurable capacity in this study, but these measurements are specific to this part of the coast (so thus may be of interest to local ecologists) but globally are almost meaningless. The error is accommodated statistically in this study, by dealing with average elevation of upper limits of each species' holdfasts. Variation in aspect and wave exposure are site specific and have been accommodated within the statistics of captured data. No pre-earthquake measured data is available by site, and even if it was, the granularity of the bouldery intertidal environment means that changes in these factors are likely to have been wrought by coseismic uplift.

Another thing that would be informative is to know what timescale this technique is available over. The authors carried out their work after a few months, but how much longer could they have realistically applied the technique. Also, is the 20 holdfasts that
seems to be their lower limit statistically valid. If the authors take their first five, ten, twenty measurements at a site, does the result change significantly. Would it improve if they used fifty? Maybe start by showing the number of measurements of each type on Figure 5. Maybe somehow on Figure 6 too.

Response – The timescale of organic decay following coseismic uplift would vary by location, time of year, and organism targeted. Also, decay of the holdfasts and remaining algal fronds was variable with aspect. In dry, sunny places, for example, the holdfasts measured were highly decayed and rudimentary, while in some places algal fronds were still present, even at the same site. Additional comments are made at lines 459-461 to incorporate the reviewer's feedback. Figure 5 adjusted to include "n" number of measurements, as a comparison to the data ranges already given. This is not done for other figures, as the figures become so cluttered as to be unreadable. Further, besides for each data point presented in Figure 5, we have now included in the figure the mean value.

One strategy for the discussion is already present in the introduction. The authors provide an extensive list of coastal uplifts and biological assessments thereof in the introduction. I have always thought that the discussion should revisit the key points of the introduction. So, please revisit that list and discuss the advantages and limitations of this technique. How many historic earthquakes have caused coastal uplift, what magnitudes of coastal uplift have been documented, using what biological indicators. Which of them could have been targeted with this technique, and over what timescale. Maybe put together a useful chart/table showing the preservation potential, accuracy, precision, ease of deployment, best vertical resolution (critical if you want to document uplifts of half a meter or less), speed of survey, skill requirements, etc of the various techniques and illustrating why this one is important – I am thinking something like Table 2B in Quigley et al. (2016), in which the lead author was responsible for another biological assessment of vertical displacement. Or maybe a McCalpin style graphic. Whatever you do, please re-read your introduction and use it to put some spice into the

discussion.

Response – This paper is presenting a new methodology, and does not seek to be a review paper, which is what is implied here. While a review that compared all methodologies for past earthquakes may be useful, it is also incredibly difficult, for these or any authors to provide realistic commentary that the reviewer is requesting. Ultimately preservation potential, vertical resolution, precision and biological indicator used are all local effects, and should be determined by local or location informed reviewers (as was the case with the lead authors role in Quigley et al 2016).

These are just a few ideas, but I really think the author team is uniquely qualified to lead this discussion towards a useful earthquake-ecology viewpoint, especially given the lead author's established expertise in biological zonation (Reid et al., 2011 - their fig 2 deserves a citation here) and earthquake effects on that in New Zealand (Quigley et al., 2016). One way or another the discussion needs to be beefed up. A brief discussion of some of these points, possibly using these results to think further about some previous Reid et al work, would probably find favour with coastal ecologists and increase the citability of the paper. A table or graphic of the kind I suggest above would surely contribute to the quantitative coastal ecological impact assessments that will follow future earthquakes, in New Zealand and elsewhere. Enough long-winded discussion of the discussion (eek). Another key criticism relates to the description of the methods. I found this opaque, with many ambiguities. The opacity is largely because the paper dives straight into formulae without really explaining the strategy. This is not helped by a major discrepancy between the formula shown in text and that shown in Figure 3. After some careful checking I am happy that the methods are valid but they need to be clarified and subscripts used consistently. I also see little indication of the uncertainties in the figures. They are covered to a certain extent in the text, but there are no error bars on major figures (e.g. Fig 6).

Response – the more detailed comments of the reviewer, that certainly improve the clarity of the methodology, have all been addressed below. Error bars are not included

on some figures (e.g. Fig 6) as this would result in cluttering and unnecessary complication of those figures. However, all data, together with their uncertainties, are presented in the tables that accompany this manuscript.

Below I provide detailed comments and corrections, mostly regarding the methods and description thereof. There may be some errors of understanding on my part and I apologise for any such errors contained here. Once again, thank you for this opportunity and good luck to the authors. I look forward to seeing this published. Sincerely, Brendan

Detailed comments

Line 25: Satellite data is ubiquitous, Lidar replacement is the real target here.

Response: Sentence updated.

Line 31: Surely Darwin could get a mention here?

Response: This is already present in more detail at lines 56-59. Graham (1824) was the initial user of this technique, rather than Darwin, and the reference to the Beagle voyage is present in Fitzroy (1839).

Line 78: Not convinced this is the right reference. What about Williams et al. (2013)?

Response: Eberhart-Phillips & Bannister 2010 comprehensively map the seismicity and Vp/Vs variations in 3 dimensions within the upper plate and slab in the Marlborough region.

Line 83: An earthquake network comprises several faults. An earthquake is a process that ruptures a fault or a network of faults. An earthquake does not comprise a network of faults.

Response: Sentence revised at lines 116 and 117.

Line 97: Vaguely and unintentionally implies that the mapped surface faults and coastal

uplift, as well as the tsunami, extended 250 km south of Kaikoura.

Response: Punctuation corrected at line 141 to remove this implication. Sentence modified, as well.

Line 102: Here and elsewhere (e.g., line 103, 105, 114. . .), the word 'exposed' and derivatives of that word are used in two different senses of the word – Exposed coastlines and exposed holdfasts. Sometimes it is clear from context and sometimes it requires a double read to figure it out. Please consider using expos. . . in one sense only and replacing the other meaning with a different word. Line 114 is particularly bad – controlled by exposure above the tide? Lack of shelter? It is not an easy issue to address but maybe keep exposed for 'unsheltered' and use qualifications such as subaerially-exposed, terrestrially-exposed, outcropping, etc for stuff that is above sea level.

Response: The text is revised at lines 146, 148, 150-151, 155, 158, 163, 187, 197, 199 and 201. This was an issue that had been addressed, but the comments of an external reader are appreciated in continuing to improve this issue. "exposed" is common terminology to describe coasts impacted by ocean swell and waves, and likewise "exposed" is used by geologists to refer to rocks in outcrop. In this manuscript exposed is retained and clarified for use with reference to coasts, and modifications or synonyms are used in geological contexts.

Line 106: It is not diurnal, it is semi-diurnal, with two full cycles daily.

Response: Corrected at line 151.

Line 188: It seems to me that you never explicitly state the tide gauge uplift except in Table 1. Why not?

Response: The results of calculations to determine uplift using the tide gauge are given in the results section. This is not included in this methods section.

Line 189: Change to "Biological data collection" and then add a new section title at line

220 - "Data processing"

Response: Change made and new heading added, and section 3 subheading numbering updated

Line 211: Were the wave effects given a plus-minus value?

Response: No. The wave effects are intrinsically variable across minutes, hours, days and seasons, and the goal was to measure water-level and the height of holdfasts above water-level. The text has not been adjusted here, as to do so adds an unnecessary complication.

Line 213: Was sea-level re-measured after each group of twenty?

Response: Yes. The text has been updated at line 306-307 to clarify this.

Lines 215-218: As somebody with building experience I would have forgotten about the tape and used either 1) a builder's laser level and a reflective staff. Measure the height of the laser level mount with a tape, then measure the height of holdfasts in all positions using a reflective staff. Laser levels are small, portable, and cheap as chips and the staff could be a stick with a high vis jacket. Any holdfast accessible with RTK could be done with a laser level, especially in the late evening, and would yield similar accuracy to RTK. 2) Even cheaper, a homemade water level, with the reservoir placed on a local high point. Engineers used commercial versions widely after the Chch earthquake to survey floor levels.

Response: We appreciate the value of laser levels, but their use during bright day-light hours is problematic; the best time for biological data collection was during mid- to low-tides commonly during high lighting conditions. The homemade level suggestion is indeed a very cheap and simple method to get data that may have been more accurate that our tape measure method, and I am sure readers of this review will appreciate it! Our goal at the time of collection was to go with a simple low-tech method with equipment that most people would already have in their possession. No changes to

text made.

Line 220: New section title here - maybe "3.3 Data Processing".

Response: Done

Line 221: These field measurements of apparent uplift. . . [No. they are field measurements of exposure above a reference tide level. Nobody would consider that to be apparent uplift because it is a time-dependent measurement] were then further processed to determine the total uplift [No. uplift is either of rock or surface – in this case both are equivalent at this moment in time, so just say surface uplift], taking into account the time of data measurement and the pre-earthquake living position of the algal holdfasts which is the difference between pre- and post-earthquake elevation of algal holdfasts [Note that position is a 3D thing and we are only interested in z, not x or y].

Response: The text is modified at lines 317-318.

Line 222: Just say "Three different corrections were used to derive surface uplift from elevation above sea level at a point in time. These were a) tide gauge calibration; b) interpolation of NIWA tide forecaster and c) interpolation of LINZ tide forecaster. Method a) calculated a correction using direct measurement of stranded algae relative to the

Response: On reviewing this section of text and the reviewers recommendations we feel changing the text as recommended would add further confusion. The correction factor (our terminology) was derived in slightly different ways, but it was only one part of the overall calculation of uplift. Uplift is known at the Kaikoura tide gauge, whereas methods b) and c) combined the tide forecaster interpolation with local knowledge of biological living zones. Although more subjective, corrections b and c are applicable to sites where no tide gauge is available.

Line 229: A factor is a number that is multiplied by another to yield a product. This is not a factor, it is a constant. There is also more than one constant.

Response: In a general sense, as is often applied in science text, this is meant as an adjustment to an equation to account for a known variation. However, in a pure mathematical sense, the reviewer's definition of factor is correct. As there is more than one constant in this study, and this value would need to be re-calculated at any other location we are not comfortable defining this as a correction constant – the term constant implies our correction value could be applied globally. It cannot, but the calculation can be. On this basis, we have revised the text to use the term 'correction' throughout.

Line 229: I suggest that you change title to "Correction a) Deriving living depth constants for target species using the Kaikoura tide gauge."

Response: It is not clear how this would improve the heading . . ...

Lines 230-250: Try and be a little kinder to those who want to reproduce this. I found it very hard to follow until I figured out what you were doing. Maybe describe in words what you are doing, rather than diving straight into the derivation of your constant. I recommend: "Surface uplift is the difference between the present, post-uplift elevation and the pre-uplift elevation (the living elevation) of the organisms holdfasts. The target species occupy slightly different living positions in the inter-tidal zone, so this new method first derives a constant living elevation (X, in m) for each target species ($XC$ – Carpophyllum; $XD$ – Durvillaea; Combined $XG$), by calculating the pre-earthquake elevation of the stranded holdfasts relative to the spring-tide mean low water level (MLWS). The constant $XC/D/G$ is calculated in three stages, using holdfasts at sheltered sites close to the Kaikoura tide gauge. First the height of the stranded holdfast above the uplifted tide gauge mount is calculated from the sum of the tide gauge height and the observed elevation of the stranded holdfast relative to sea level (both at measurement time). Secondly, the tide gauge uplift is subtracted obtain the elevation relative to the pre-quake tide gauge. Finally, MLWS is further subtracted to obtain elevation relative to MLWS, which is a key reference level for the biological zonation. This procedure is given by the equation: ... where

Response: The text has been modified starting at line 327 to improve clarity. The text is kept at the calculation of the correction, rather than including reference to surface uplift as suggested by the reviewer, as this seemed to remove the focus from the immediate topic. We are grateful to the reviewer for this suggestion, and the time taken to suggest suitable text for an external audience.

Line 231: Depth implies below sea level. At least one species lived above MLWS, so depth would be negative. Just use elevation, since you are differencing with elevation.

Response: Living-depth was used in reference to the tidal cycle of MLWS to mean high water spring. However as we calculate the living "depth" with respect to MLWS, this is changed to living position throughout the manuscript.

Line 235: Bit of a mess here. Outer parentheses are redundant. The MLWS subtraction is not shown in the equation on Figure 3. Figure 3 uses HTG instead of H, so please be consistent.

Response: Equation in text tidied up, and MLWS re-inserted into Figure 3. Subscripts are checked and corrected throughout the manuscript.

Line 240: subscripts rather than indices?

Response: corrected

Line 241: . . . average post-uplift tide gauge reading. . .

Response: Adjusted

Line 244: You have already said they occupy different levels, especially if you adopt the text above, so just leave out the first sentence of this paragraph.

Response: The first sentence is retained, as this text is intended to clarify the use of C/D/G with respect to X, and why these need to exist. The text suggestion for line 230 was not adopted verbatim.

Line 247: "Further, only . . .." – This should appear within the derivation of the formula. See comment for line 230 above.

Response: Adjusted

Line 256: You introduce a parameter without defining it (UB(TG)). Sure I can work out what it is but I shouldn't need to. Also H in eqn 2 looks a lot like H in eqn 1. Hence the need to use HTG in eqn 1. Each time you use H it refers to a specific time and place. I wonder if you need to have HTG and HSS (survey site). One way or another your subscripts need to be unique, informative and consistent, because at the moment they are not. I recommend that you include a glossary of the terms used in your equations.

Response: UB(TG) is defined. H is indeed HTG and this is adjusted.

Line 265: Please clarify this process. Maybe change text beginning line 265 to read... "where OMCDG is the observed elevation of the holdfasts relative to locally measured sea level, HNIWA/LINZ is the difference between the predicted tide height at survey time and the MLWS based on one year of predictive tables, and XC/D relies on expert assessment as follows: Carpophyllum..." Note here, please be consistent with subscripts. I cant see any reason why OM here would be different to OMC/D/G in equation 1. Also, how would you get an expert assessment of tidal zonation in Timor for instance?

Response: Text is adjusted, although not quite as recommended above (HNiwa is not linked to MLWS as written above). Subscripts have been reviewed and adjusted throughout. Working in other locations would require talking to local marine biologists, or local fisherman to understand their observations of local conditions.

Line 272: What is a regional height?

Response: Sentence adjusted for clarity. Refers to Kaik■ura region.

Line 274: Replace height with holdfast elevation (height is a vague term)

Response: Change made

Line 285: Why are NIWA and LINZ different? Can you illustrate the difference in a figure, or summarize it somehow?

Response: Text is adjusted at lines 428-429 to accommodate this. The LINZ charts provide data for fixed geographical locations. The NIWA forecaster uses a model to predict tidal height at any geographical point, between the LINZ fixed points.

Line 292: The absolute accuracy of RTK may be 5 cm vertical if you put your base station on a suitable order trig. The internal relative accuracy is better than that (2 cm) with favourable GPS environment (see e.g table 1 in Duffy et al. 2013). On that note, where was your base station set up? Maybe it doesn't matter, if all of your RTK measurements were differenced with a local sea level measurement every 10 minutes or so but you should still mention it. Personally, I would have opened and closed the survey at the tide gauge, so that I could see how well my sea level measurements replicated the tide gauge.

Response: A base station was not used in this case, as the internal accuracy of the difference between holdfast elevation and waterline elevation was the key data required. Absolute measurements were not taken, and would have required the more time consuming use of a base station and calibration. A minor change made in the methods section to clarify no base station was used.

Line 314: Please check figure order. I haven't checked super carefully, but Fig 7 seems to come before 5 and 6. Please make sure they are numbered in order.

Response: Reference to figure 7 at line 330 is removed, and reference to Fig. 7 is only made in the results section, as it displays results.

Line 320: The problem with the lidar at Kaikoura harbour should be mentioned here, not in the results. And really, I want to know how inconsistent they were. If you do a histogram over a couple of roads around the harbour, what do you get? Is it consistent

with the strong motion instrument or not?

Response: Thanks to the reviewer for asking this question. In answer, differential LiDAR data was not available immediately adjacent to the KaikÅ■ura Harbour site. Extrapolation from other regions produced variable results, with none able to be realistically used, or any source able to be justified over another. The text is modified at line 482 and line 576.

Line 336: New heading - 4.1 Tide gauge locality

Response: New heading inserted at line 506

Line 339: Delete sentence - "Uplift estimates derived..."

Response: Sentence deleted.

Line 344: ... tide gauge (Figure 5) and compared with uplift of the Kaikoura tide gauge (calculated in section 3.1)

Response: Text adjusted.

Line 370: New section title here - maybe "4.2 Paia Point and Omihi Point".

Response: New heading inserted at line 552.

Line 392: How inconsistent? And why? Differential beach gravel compaction? Something else? Does this inconsistentcy affect have any effect over the distance from strong motion sensor to measurement sites?

Response: This is resolved by the adjustment to the methods text as suggested above, and is repeated at line 576 for clarity.

Line 432: Basically repeats something you have said in 360 and in 226. Just say it once, with maximum impact.

Response: Left in to re-iterate point

FIGURE 2: In the caption, mention green ulva (from line 120)

Response: Figure caption adjusted to include reference to Ulva.

FIGURE 3: Fix the missing bit of the formula. Shift the Rtk data panel somewhere else or at least change the background colour, because as drawn it looks like part of the conceptual panel.

Response: Formula text box adjusted to reveal full formula. Thanks to the reviewer for the comment regarding the RTK panel. This is now changed to an alternate colour to remove confusion.

FIGURE 4: Not really correct. Why is an equation shown on Figure 3 but not on Figure 4? Surely XC/D_NIWA_LINZ is measured from MLWS??

Response: Figure 4 is correct. XC/D_NIWA_LINZ is estimated, not measured.

FIGURE 6: Caption - note that no lidar comparison was produced for Kaikoura harbour. Even better, show the real picture with appropriate error bars.

Response: Discussed above, no data was available

FIGURE 7: part b - No Rainbow, not now, not ever. Use a proper colour stretch, and stretch it over the elevation range from sea to just above the road. I don't particularly care about the hill.

Response: Clearly this is a personal preference. There are no standards for colour ramps and we feel comfortable with the Figure as it stands.

References: Quigley, M.C., Hughes, M. W., Bradley, B.A., Ballegooy, S.v., Reid, C., Morgenroth, J., Horton, T., Duffy, B., and Pettinga, J.R.: The 2010-2012 Canterbury earthquake sequence: Environmental effects, seismic triggering thresholds and geologic legacy, Tectonophysics, 672-673, 228-274, 2016.

Reid, C. M., James, N. P., and Bone, Y.: Carbonate sediments in a cool-water

macroalgal environment, Kaikoura, New Zealand, Sedimentology, 58, 1935-1952, 2011.

Please also note the supplement to this comment:
https://www.earth-surf-dynam-discuss.net/esurf-2019-46/esurf-2019-46-AC1-supplement.pdf

———————————————————
[Figure]

Pre-EQ MHWS = 4.05m (TG) — Post-EQ MHWS = 3.1m (TG)

Pre-EQ MHWN = 3.7m (TG) — Post-EQ MHWN = 2.75m (TG)

**Tidal zone**

Stranded holdfasts

$OM_{C/D/G}$

Living holdfasts

Pre-EQ MLWN = 2.5m (TG) — Post-EQ MLWN = 1.5m (TG)

$X_{PRE}$

Pre-EQ MLWS = 2.05m (TG) — Post-EQ MLWS = 1.1m (TG)

$H_{TG}$

**KAIKOURA EARTHQUAKE**

$X_{C/D/G} = ((H_{TG}+OM_{C/D/G})-U_{TG})-MLWS$

Tide Gauge (TG) basal rock mount and attached
holdfasts uplifted by 0.96m ($U_{TG}$)

**MLWS**

All data $X_D$

Carpophyllum $X_C$

Durvillaea $X_D$

$X_{CDG}$ offset from MLWS (m)

**RTK data**

**KAIT $U_{TG}$ = 0.96m**

**Fig. 1.**

[Figure]

RTK data (red points, shaded orange region):
- All data: n=69, 0.97
- Carpophyllum sheltered: n=38, 0.96
- Durvillaea sheltered: n=31, 0.98

Tape measure data (black points, shaded grey region):
- All data: n=60, 1.05
- Carpophyllum sheltered: n=36, 0.98
- Durvillaea sheltered: n=14, 1.10
- Durvillaea exposed: n=10, 1.21

Uplift measured in Tide Gauge (0.96m) — solid line
Uplift from diff. LiDAR (0.92m) — dashed line

**Fig. 2.**

---

## Author Comment (AC2) · 19 Mar 2020

Response to the Anonymous Reviewer-2

Important note: our responses to the Anonymous Reviewer's comments are in red italics and the line numbers in our responses refer to the revised version of the manuscript.

Dear editor,

Thank you for giving the opportunity to review the paper by Reid et al.

I read with interest the submitted manuscript and overall I believe it is an interesting contribution fitting the scope of the journal. The authors use tidal data and the eco-

logical preference of intertidal algae to determine coastal uplift. They calibrate their biological data with both real-time and predictive tidal charts and compare their results with lidar and strong motion data. Overall, I think their proposed methodology will be of interest to many coastal geomorphologists and geologists working with coastal deformation and their paper deserves publication.

We thank the reviewer for the positive comments.

However, in order for their methodology to be replicated some revisions are necessary, primarily in the methods section. In particular, paragraphs 3.2.1 – 3.2.3 are quite hard for a reader to follow and more complicated that necessary. I suggest the authors to improve this part by explaining in a better way their methodological approach.

The manuscript has been clarified accordingly as a response to both reviewers' comments.

Some technical corrections: line 376: include the error bar –> error is provided at line 559 line 389-390: error bar for NIWA and LINZ? –>error is provided at lines 574 and 575 line 418: middle on line, "they" is not needed–> Done. Caption of figure 1: line 570, note that the position of State Highway One is the yellow line –> Corrected. Figure 6: I would suggest to include error bars –> Errors are presented in Table 5 as they are too small to be represented graphically.

Please also note the supplement to this comment:
https://www.earth-surf-dynam-discuss.net/esurf-2019-46/esurf-2019-46-AC2-supplement.pdf